# Exploration of Preventable Hospitalizations for Colorectal Cancer with the National Cancer Control Program in Taiwan

**DOI:** 10.3390/ijerph18179327

**Published:** 2021-09-03

**Authors:** Yu-Han Hung, Yu-Chieh Chung, Pi-Yueh Lee, Hao-Yun Kao

**Affiliations:** 1Department of Nursing, Yuan’s General Hospital, Kaohsiung 80249, Taiwan; yuhan.horng@gmail.com (Y.-H.H.); y0093@yuanhosp.com.tw (P.-Y.L.); 2Department of Healthcare Administration and Medical Informatics, College of Health Sciences, Kaohsiung Medical University, Kaohsiung 80708, Taiwan; jeff0102034@gmail.com; 3Department of Medical Research, Kaohsiung Medical University Hospital, Kaohsiung 80708, Taiwan

**Keywords:** preventable hospitalizations, ambulatory care sensitive conditions, colorectal cancer, Poisson regression model

## Abstract

Background: Causing more than 40,000 deaths each year, cancer is one of the leading causes of mortality and preventable hospitalizations (PH) in Taiwan. To reduce the incidence and severity of cancer, the National Cancer Control Program (NCCP) includes screening for various types of cancer. A cohort study was conducted to explore the long-term trends in PH/person-years following NCCP intervention from 1997 to 2013. Methods: Trend analysis was carried out for long-term hospitalization. The Poisson regression model was used to compare PH/person-years before (1997–2004) and after intervention (2005–2013), and to explore the impact of policy intervention. Results: The policy response reduced 26% for the risk of hospitalization; in terms of comorbidity, each additional point increased the risk of hospitalization by 2.15 times. The risk of hospitalization doubled for each 10-year increase but was not statistically significant. Trend analysis validates changes in the number of hospitalizations/person-years in 2005. Conclusions: PH is adopted as an indicator for monitoring primary care quality, providing governments with a useful reference for which to gauge the adequacy, accessibility, and quality of health care. Differences in PH rates between rural and urban areas can also be used as a reference for achieving equitable distribution of medical resources.

## 1. Introduction

Causing more than 40,000 deaths each year, cancer is one of the leading causes of mortality in Taiwan, resulting in great suffering and expense to families and the national health care system. To reduce the incidence and severity of cancer, the National Cancer Control Program (NCCP) has increased the rate of screening for various types of cancer [1,2]. Initiatives have also begun to improve accessibility to primary care, reduce preventable hospitalizations (PH), reduce the growth of medical expenses, and increase cost efficiency [3,4,5,6].

In the 1980s, the United States (US) national healthcare policy adopted improvements to access primary care as an important indicator for the overall effectiveness of the health care system. One measure towards this was the introduction of “ambulatory care sensitive conditions (ACSC)” [7,8,9], a concept in which some diseases may be prevented through appropriate and immediate primary care, thus minimizing PH through immediate and effective outpatient care. Past studies have noted that PH is an effective indicator of the accessibility, quality, and performance of primary care, and can help health care providers better understand the effectiveness of medical intervention [10,11]. Communities with poor access to medical care suffer higher rates of hospitalization for chronic disease, and ACSC can serve as a key factor to assess improvements in access to care by reducing PH [12,13,14].

Research in the early 1990s focused on the accessibility of primary care, and sought to control costs through defining comparison tables for sensitive outpatient care status [7,15]. In 1993, the US Institute of Medicine (IOM) proposed official standards for immediate and efficient primary care, seeking to minimize incidence of disease, control the deterioration of acute conditions, manage chronic disease, prevent complications, and prevent hospitalizations [5,8,16]. The IOM also proposed using PH as an indicator for the availability and quality of primary care, suggesting that hospitalizations caused by ACSCs indicate non-optimal primary care quality and access. The Health Care Research and Quality Service of the US Department of Health and Human Services (DHSS) defines PH as hospitalizations caused by ACSCs. Many other countries have begun to develop ACSC scales adapted to local conditions [5,17].

Several studies have shown that health care system organization also has an impact on PH rates. Bindman et al. (2005) pointed out that the reimbursement system of the US Medicaid managed care scheme can reduce the PH rate more effectively than the Medicaid fee-for-service scheme [18]. Schreiber et al. (1997) also found that hospitals with higher rates of inpatient remuneration tend to have higher PH rates [19,20]. Zeng et al. (2006) found that individuals in Medicare HMOs have lower rates of hospitalization than Medicare fee-for-service beneficiaries [21]. In a preliminary cross-sectional pilot study in Catalonia, Spain, Starfield (2001) suggested that ACSC is a good indicator of primary care quality in terms of hospitalization avoidance [22]. As markers of primary care effectiveness, ACSC identifies health conditions conducive to specific aspects of primary care by minimizing variation of hospital admission policies. An expanded list should be useful for evaluating global performance of primary care and for analyzing market response for primary and specialty care [23].

In a cross-sectional study, Wolff et al. (2002) randomly sampled over 1.2 million US senior citizens registered with Medicare Part A or Part B, finding that individuals with multiple chronic conditions were increasingly likely to incur inpatient admissions for ACSCs and to incur preventable complications during hospitalization [24]. Better primary care, especially better care coordination, could reduce PH rates, particularly for individuals with multiple chronic conditions [25]. Many studies have examined the relationship between primary care availability and PH. For example, hospitalization due to ACSCs is often used as an indicator to measure access to primary care [26]. Elderly patients residing in rural Nebraska, an area characterized by health professional shortages, were found to incur fewer hospitalizations due to chronic ACSCs if they had access to at least one rural health clinic. A number of other studies have shown similar results. Cloutier-Fisher et al. (2006) suggest that access to effective primary care in rural communities remains problematic in British Columbia based on failure to reduce PH rates [27]. Continued monitoring of discrepancies in hospitalization rates remains important as a reflection of inequity in service provision, particularly in terms of the relative needs of rural and urban populations [28]. Laditka et al. (2009) suggest that rural areas are more prone to incidence of ACSCs, suggesting urban/rural disparities in access to primary health care [29].

In 2005, Taiwan launched a national cancer prevention and treatment program, including continuous cancer surveillance and increased funding for primary care. However, numerous scientific studies have shown that the implementation of universal single-payer health insurance is not necessarily conducive to the balanced development of primary care. Hospitalization for ACSCs is an accepted indicator of access to health care and avoidable morbidity [26]. Accessible care of reasonable quality should reduce incidence of ACSCs. Little research has examined the associated with external factors or interventions, such as NCCP [30,31]. To date, no research has examined the relationship of such interventions (e.g., National Cancer Control Program) on minimizing preventable hospitalizations of colorectal cancer in Taiwan [32,33]. Therefore, the primary motivation behind this study is to determine whether the implementation of Taiwan’s NCCP has contributed to a long-term decline in PH rates, and to identify factors which may reduce PH. A cohort study has been conducted to explore the long-term trends in PH/person-years following NCCP intervention from 1997 to 2013.

## 2. Materials and Methods

Trend analysis was conducted based on patient characteristics (age, gender, comorbidities, and insurance premium) and medical care quality characteristics (total number of hospitalizations, medical institution level, and medical institution area) for long-term hospitalization. The Poisson regression model was used to compare PH/person-years before (1997–2004) and after intervention (2005–2013), and to explore the impact of policy intervention in reducing hospitalization rates. Because the dependent variables of this study were discrete (counting) data, suitable for the use of Poisson regression analysis, they do not require the use of residual analysis. The residual analysis indicates that the dependent variables are continuous and are used under the assumption of normal distribution; therefore, it is not applicable in the present analysis.

The study received IRB approval from Kaohsiung Medical University (IRBKMUHIRB-EXEMPT(I)-20190004) on 13 February 2019. Data were taken from Taiwan’s national health insurance research database published by the National Institutes of Health (NHRI) in 2000. Research subjects were obtained from the Outpatient Prescriptions and Treatment Details File (CD) and Inpatient Medical Expenses List (DD) from 1997 to 2013. Patient characteristics data were obtained from the claim data for reimbursement from the 1997 to 2013 data file (ID) along with the registry for contracted medical facilities (HOSB) from 1997 to 2013. PH is defined in reference to prevention quality indicators (PQIs) proposed by the US Agency for Health Care Research and Quality (AHRQ) in 2014 [14,34,35], and applied to the National Health Insurance database to identify patients with colorectal cancer from 1997 to 2013, resulting in 8051 individuals for whom complete data are available. The sample selection scheme is summarized in Figure 1.

This study describes patient characteristics (age, gender, comorbidity, and insurance premium) and medical care quality (total number of hospitalizations, medical institution area, and medical institution level) of the study subjects in terms of frequency and percentages. Annual statistics and trend analysis for PH are also presented at the various stages of the national cancer prevention and control plan implementation.

This study used trend analysis to investigate changes in the ratio of hospitalization per person-year at different points in time (1997 to 2013), thus determining the timing of the effect of NCCP. The bivariate analysis used a chi-square test (χ² test) to explore the correlation between age, gender, insured amount, total hospitalizations, comorbidities, medical institution level, medical institution area, and hospitalization avoidance.

The Poisson regression model was used to analyze PH/person-years prior to (1997–2004) and after intervention (2005–2013), and to explore the impact of strategic interventions on hospitalization avoidance in terms of hospitalizations/person-years.

## 3. Results

This study is based on data collected for 8051 patients from 1997 to 2013. Slightly more than half of the sample (54.7%) was male. The average age of the total sample was 61.7 years old, with an average score of 3.5 on the Charlson comorbidity index (CCI). The CCI comprises 19 comorbid conditions, where each disease is given a different weight based on the strength of its association with 1-year mortality (Charlson et al., 1987). Among the sample, 4167 patients (51.8%) were classified as high comorbidity (CCI > 8), followed by 1600 (19.9%) as medium (5 < CCI ≤ 8). In terms of insurance premiums, the sample is divided into six classifications: 2697 (33.5%) were classified as dependents because they had no fixed income, followed by 2149 individuals (26.7%) who paid premiums between USD 620 and USD 815. Descriptive statistics for the remainder of the samples are presented in Table 1.

Among the sample, the average number of PH (person-time/person-year) was 1.1. In terms of geographical location, 4204 (52.2%) of PH incidents occurred in hospitals in Taiwan’s north region, followed by the south (27.3%) and central regions (18.2%). Most incidents (60.0%) occurred in major medical centers, followed by regional hospitals (13.5%). The chi-square test (χ² test) was used to explore the correlation between gender, age, comorbid conditions, insurance premium, total number of hospitalizations, medical institution type, region, and hospitalization avoidance. The results indicate that gender, comorbidity, insurance premium, region, and medical facility type are all important factors in hospitalization prevention (*p* < 0.0001). Analytical findings are summarized in Table 1.

The Poisson regression model was used to analyze the PH/person-years before (1997–2004) and after intervention (2005–2013), and explore the effect of policy intervention on hospitalizations/person-years. The detailed results are shown in Table 2. We found that, following the policy intervention, hospitalization risk fell by 26%, which was statistically significant at *p* < 0.0001. In terms of gender, men’s hospitalization risk was 1.15 times that of women, which is also statistically significant (*p* < 0.0001). In terms of age, the risk of hospitalization doubled with every additional 10 years of age, but this was not statistically significant (*p* = 0.98). In terms of comorbidity, for every additional point, the risk of hospitalization increased by 2.15 times, which was statistically significant (*p* < 0.0001). In addition, the risk of hospital admission doubled for each 10-year increase, which was not statistically significant (*p* = 0.98).

In terms of insurance premiums, the hospitalization risk of the dependent population was 0.95 times that of those with premiums less than USD 620, which is statistically significant (*p* = 0.001). Although statistically significant, the increased risk of PH for disadvantaged groups was 0.95 times that of the minimum basic salary group (USD 620), which is not consistent with previous research findings. In addition, the remaining insurance premium groups were compared with the minimum basic salary group, and the risk ratios were all less than 1.0, which is roughly similar to the results of past studies. Detailed results are shown in Table 2.

In terms of medical institution type, hospitalization risk at district hospitals was 1.06 times that of the primary clinics, which was statistically significant (*p* = 0.006). The risk of hospitalization for the population attending regional hospitals was 0.99 times the risk of hospitalization for the population attending primary clinics, which was not statistically significant (*p* = 0.473). Hospitalization risk at major medical centers was 0.83 that of primary clinics, which was statistically significant (*p* < 0.0001).

Trend analysis was used to verify that different time points (1997–2013) can validate changes in the number of hospitalizations/person-years. Detailed trend curves are summarized in Figure 2.

## 4. Discussion

The results of this study show that patient age and medical facility type are not significant factors following NCCP intervention. Gender, comorbidity, insurance premium, region, and medical facility type are significantly related to PH, both before and after the NCCP intervention. Previous studies have found that PH rates increase with age, but the previous study did not find patient age to be a significant factor in PH rates. In this study, the PH risk increases with age because this study only included colorectal cancer patients with an average age of 61.7 years. Future research should explore the effect of age on different diseases.

Consistent with previous studies, this study finds that men are at increased risk for PH. For those with single comorbidities, previous studies have also found that men with diabetes have a higher risk of PH compared to women with diabetes. The present study found that patients with higher comorbidity scores were at significantly greater PH risk, with hospitalization risk increasing by 2.15 with each 1-point increase in CCI score, consistent with Shan’s (2004) findings that chronic disease and poor general health status significantly increases PH risk.

Previous studies have shown that low-income people have a greater need for health care. Since this research has not been able to obtain the actual salary income of the research subjects, the insurance premium of the research subject was used as a substitute for income, but this may underestimate actual income level. The results indicate that hospitalization risk is greatly reduced as the insured amount increases, consistent with previous findings. In addition, risk of hospitalization was found to increase with distance from urban areas, indicating that health care outcomes are associated with accessibility of medical resources, and that additional government effort is needed to further address the urban–rural gap for medical resources. Type of medical institution is also found to have a significant association with hospitalization, with local clinics and regional hospitals (i.e., institutions with relatively limited resources) correlating with increased hospitalization rates.

The Poisson regression analysis found that the risk of hospitalization after intervention was about 24% less than before intervention, indicating the National Cancer Prevention Program (NCPP) effectively reduced PH rates. However, previous research in Taiwan found that factors such as copayments and pay-for-performance for specific diseases can reduce PH. International studies have also found PH rates can be reduced by medical policy interventions such as managed care or affordable care. The current findings reflect increased PH rates up to 2004, but they began decreasing in 2005 with the policy implementation, presenting an inverted V-shaped trend. A similar study of the turning point is also consistent with the outcomes of associated projects. For example, a generational study by Taiwan’s National Science Council found a V-shaped trend in terms of age and gender on the PH rate, with the inflection point occurring in 2005. To control national insurance expenditures, the NHI first implemented the fee-for-service payment system, followed by the implementation of global budgets for hospitals in 2002. The introduction of cancer screening programs had a clear effect on overall insurance expenditures. At present, few studies in Taiwan have examined the effect of NCCP on PH. The main findings of this study focus on colorectal cancer, which has a relatively small sample size; thus, changes in health care payment systems may also contribute to the PH turning point in 2005. Future work will include other indicators such as AHRQ.

Based on our results, this study presents the following recommendations: PH should be more broadly adopted as an indicator for monitoring primary care quality, providing governments with a useful reference by which to gauge the adequacy, accessibility, and quality of health care. Differences in PH rates between rural and urban areas can also be used as a reference for achieving equitable distribution of medical resources.

## 5. Conclusions

The present study is subject to certain limitations. Clinical disease identification codes vary among national health care systems; thus, the recorded disease code may not match the actual diagnosis. Previous studies have generally found that socio-economic conditions, marital status, education, and residency all affect PH, but the present study was not able to directly measure socio-economic status. Moreover, this longitudinal study may be too optimistic in terms of incidence of PH. Future work could seek alternatives, such as using cross-sectional research designs to verify the relationship between PH and colorectal cancer to reduce the impact of time series. Finally, we suggest that future researchers could investigate other PH indicators to provide more insight into other factors that are associated with PH for specific diseases.

## Figures and Tables

**Figure 1 ijerph-18-09327-f001:**
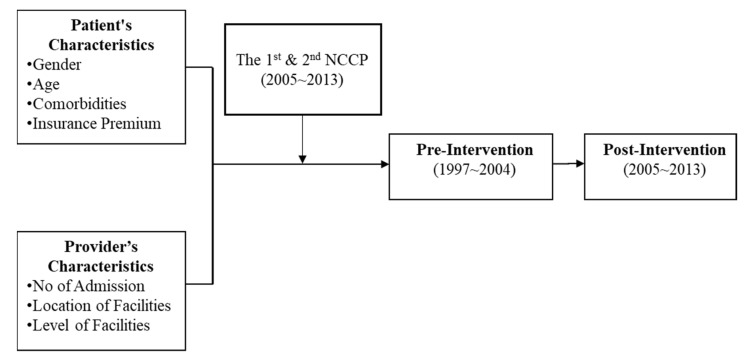
Sample selection scheme.

**Figure 2 ijerph-18-09327-f002:**
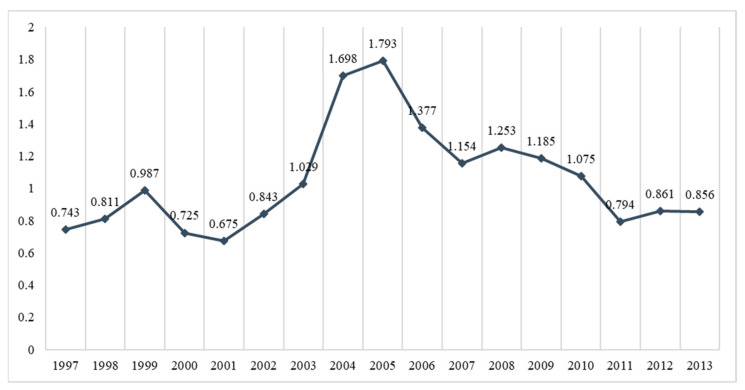
The trend of preventable hospitalizations (person no./person year).

**Table 1 ijerph-18-09327-t001:** Descriptive analysis.

Variables	Pre—(1997–2004) *N* = 3990		Post—(2005–2013) *N* = 4061		Total—(1997–2013) *N* = 8051	
	*N* (%)	*p*	*N* (%)	*p*	*N* (%)	*p*
Patient’s Characteristics						
Gender		<0.0001		<0.0001		<0.0001
Male	2149 (53.9%)		2255 (55.5%)		4404 (54.7%)	
Female	1841 (46.1%)		1806 (44.5%)		3647 (45.3%)	
Age		0.442		0.068		0.746
MEAN ± SD	61.43 ± 9.53		62.09 ± 9.88		61.76 ± 9.71	
CCI		<0.0001		<0.0001		<0.0001
MEAN ± SD	3.68 ± 1.81		3.37 ± 1.88		3.52 ± 1.86	
None	496 (12.4%)		593 (14.6%)		1089 (13.5%)	
Low (<5)	500 (12.5%)		695 (17.1%)		1195 (14.8%)	
Median	737 (18.5%)		863 (21.3%)		1600 (19.9%)	
High (>8)	2257 (56.6%)		1910 (47.0%)		4167 (51.8%)	
Premium/Month		<0.0001		<0.0001		<0.0001
Dependency	1369 (34.3%)		1328 (32.7%)		2697 (33.5%)	
<620 USD	867 (21.7%)		783 (19.3%)		1650 (20.5%)	
620–760 USD	1084 (27.2%)		1065 (26.2%)		2149 (26.7%)	
761–1210 USD	252 (6.3%)		312 (7.7%)		564 (7.0%)	
1211–1526 USD	209 (5.2%)		283 (7.0%)		492 (6.1%)	
>1526 USD	209 (5.2%)		290 (7.1%)		499 (6.2%)	
Provider’s Characteristics						
No of PH (Person No/Year)	0.939 ± 2.449		1.151 ± 5.815		1.051 ± 4.231	
Location		<0.0001		<0.0001		<0.0001
North	2073 (52.0%)		2131 (52.5%)		4204 (52.2%)	
West	750 (18.8%)		713 (17.6%)		1463 (18.2%)	
South	1058 (26.5%)		1140 (28.1%)		2198 (27.3%)	
East	81 (2.0%)		65 (1.6%)		146 (1.8%)	
Off-Island	28 (0.7%)		12 (0.3%)		40 (0.5%)	
Level		<0.0001		0.061		<0.0001
Medical Center	2066 (51.8%)		2766 (68.1%)		4832 (60.0%)	
Regional Hospital	647 (16.2%)		429 (10.6%)		1076 (13.4%)	
District Hospital	596 (14.9%)		489 (12.0%)		1085 (13.5%)	
Clinics	681 (17.1%)		377 (9.3%)		1058 (13.1%)	

**Table 2 ijerph-18-09327-t002:** The results of Poisson regression.

Variables		No of PH (Person No/Person Year)			
		Total (1997–2013)			
		RR	95% CI		*p*
			Lower	Upper	
Pre vs. Post	Pre (Baseline)	-	-	-	-
	Post	0.739	0.722	0.757	<0.0001
Patient’s Characteristics					
Gender	Female (Baseline) Male	-	-	-	-
		1.146	1.119	1.173	<0.0001
Age	Every 10 years old	1.005	0.988	1.012	0.977
Comorbidities (CCI)	Every 1 score	2.152	2.116	2.188	<0.0001
Insurance Premium	<USD 620 (Baseline)	-	-	-	-
	Dependency	0.945	0.918	0.973	0.001
	USD 620–760	0.868	0.841	0.897	<0.0001
	USD 761–1210	0.765	0.720	0.813	<0.0001
	USD 1211–1526	0.721	0.672	0.773	<0.0001
	>USD 1526	0.725	0.673	0.780	<0.0001
Provider’s Characteristics					
Location	North (Baseline)	-	-	-	-
	West	1.182	1.147	1.219	<0.0001
	South	1.122	1.092	1.152	<0.0001
	East	1.196	1.104	1.296	<0.0001
	Off-Island	1.150	1.061	0.914	0.436
Level	Med Center (Baseline)	-	-	-	-
	Reginal Hospital	1.057	1.016	1.099	0.006
	District Hospital	0.985	0.946	1.026	0.473
	Clinics	0.834	0.807	0.862	<0.0001

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
