# Peer review of "Exploration of Preventable Hospitalizations for Colorectal Cancer with the National Cancer Control Program in Taiwan"

_ijerph, 2021, doi:10.3390/ijerph18179327_

Round 1
Reviewer 1 Report
Overall this is a top quality research paper with large data set and solid analytical method. Only minor revision is suggested for some grammatical errors such as past tense should be used to describe the work that has been completed (in Materials and Methods).
Author Response
Thank you for this positive feedback. We used the editing services of Publisher to correct some grammatical errors and make the meaning clear. We have provided the English-Editing-Certificate for your reference. Please see the attachment.

Reviewer 2 Report
This manuscript is well written and addresses a critical topic. The literature review is extensive and the results are clearly reported. I would like to have seen a more detailed description of the data analysis procedures. Specifically, I think it would be helpful to provide a brief theoretical description of the Poisson regression model and explain why it is appropriate in this scenario.
In reporting the chi-square test results I suggest avoiding words that suggest causality (e.g. "factors in hospitalization prevention"). This test only provides evidence that the variables are associated. I also suggest reporting the standardized residuals in Table 1.
In the discussion section, I suggest avoiding words such as "influence", "impact", etc., which suggest causal relationships. The analyses used in this study provide evidence of association and prediction but do not support causal inferences. For instance, on line 214-215 the phrase" to have a significant impact" can be changed to "predicts" or "is associated with".
